# A Study on Dynamic Motion Planning for Autonomous Vehicles Based on Nonlinear Vehicle Model

**DOI:** 10.3390/s23010443

**Published:** 2022-12-31

**Authors:** Xin Tang, Boyuan Li, Haiping Du

**Affiliations:** 1Fok Ying Tung Research Institute, Hong Kong University of Science and Technology (HKUST), Guangzhou 511458, China; 2Research Centre for Intelligent Transportation, Zhejiang Lab., Hangzhou 311000, China; 3Faculty of Engineering and Information Science, University of Wollongong, Wollongong, NSW 2522, Australia

**Keywords:** obstacle avoidance, path planning, Takagi–Sugeno fuzzy model, autonomous vehicle

## Abstract

Autonomous driving technology, especially motion planning and the trajectory tracking method, is the foundation of an intelligent interconnected vehicle, which needs to be improved urgently. Currently, research on path planning methods has improved, but few of the current studies consider the vehicle’s nonlinear characteristics in the reference model, due to the heavy computational effort. At present, most of the algorithms are designed by a linear vehicle model in order to achieve the real-time performance at the cost of lost accuracy. To achieve a better performance, the dynamics and kinematics characteristics of the vehicle must be simulated, and real-time computing ensured at the same time. In this article, a Takagi–Sugeno fuzzy-model-based closed-loop rapidly exploring random tree algorithm with on-line re-planning process is applied to build the motion planner, which effectively improves the vehicle performance of dynamic obstacle avoidance, and plans the local obstacle avoidance path in line with the dynamic characteristics of the vehicle. A nonlinear vehicle model is integrated into the motion planner design directly. For fast local path planning mission, the Takagi–Sugeno fuzzy modelling method is applied to the modeling process in the planner design, so that the vehicle state can be directly utilized into the path planner to create a feasible path in real-time. The performance of the planner was evaluated by numerical simulation. The results demonstrate that the proposed motion planner can effectively generate a reference trajectory that guarantees driving efficiency with a lower re-planning rate.

## 1. Introduction

The path planning and path-tracking control strategies have attracted focused attention in recent years. The design of path-tracking control usually considers the detailed vehicle model to improve the control performance [1,2,3]. Particularly, a adaptive path-tracking control approach is proposed based on the lateral dynamics model with uncertain vehicle parameters [1]. Furthermore, a nonlinear sliding mode controller is proposed for the path-tracking control of an unmanned agricultural tractor by considering the kinematic model with wheel slip [2]. Yao et al. design a deep reinforcement training approach to learn the vehicle dynamics behaviour during the path-tracking [3]. However, considering the vehicle dynamics model in the path planning strategy design has been less addressed in the literature and will be the focused topic in our study.

The random search method based on vehicle dynamics model has been widely applied to conduct local path planning [4,5,6]. However, most of them are established based on a linear dynamics model to reduce the difficulty of controller design. The path planner using a complex nonlinear vehicle dynamics model [7,8] is still a big gap in the research. In order to optimize the speed of planning and convergence, and the stability of the fuzzy neural network, Xiong et al. [9] presented a fuzzy neural network controller based on a Takagi–Sugeno (T–S) fuzzy model with simple membership function in the local obstacle avoidance path planning for autonomous mobile robots. However, the disadvantage of this type of algorithm is the planning of the obstacle avoidance path without consideration of the actual vehicle dynamics model. In vehicle attitude control, a high-speed vehicle in a complex dynamic environment has a limited degree of freedom compared to a low-speed robot with a high degree of freedom for rotation [10]. For this reason, it is difficult for an autonomous vehicle to avoid the dynamic obstacles quickly and ensure the stability of the vehicle dynamics simultaneously without considering the specific vehicle dynamic characteristics. More recently, Chen et al. presented an algorithm to optimize the path-planning performance based on the fourth-order Bezier curve [11]. However, the optimization of the feasible trajectory set in this research just considers the lateral slip of the vehicle body without the consideration of a vehicle dynamics model and constraints of vehicle actuators. Shiller and Gwo [12] presented a planner integrated with a single point dynamics model to analyze the alternative paths, and then the planner chose the feasible path selected from the ones that the vehicle does not slip on when steering. However, in this algorithm the trajectory generation should conduct the forward simulation on the vehicle’s dynamic model continuously, which makes it hard to meet the real-time control requirement.

On the other hand, choosing an appropriate algorithm makes the planning results faster and more feasible. The traditional dynamic path-planning methods, such as the RRT algorithm [13,14] and road grid search algorithm [15], conduct a random search program to create a series of alternative paths in advance, and to compare the paths one by one to select the optimal path. In order to improve controller performance in real-time [16], and to make it more robust [17], some researchers are trying to optimize the method of random search in dynamic path planning. Frazzoli and Dahleh proposed an optimal path search method [18], which applies a closed-loop planning process (to compare possible local target points by random sampling method with the optimal cost function) instead of an open-loop calculation process (to conduct random sampling in existing routes with multiple iterations). In [19], Kuwata and Jonathan further proposed an RRT algorithm integrated with a closed-loop forward simulation to conduct real-time dynamic path planning. It is noted in the feasibility analysis of the optional path that the planner effectively considers the constraints of vehicle rollover limit and the safety zone limit for obstacle avoidance. With this preliminary consideration, the feasible obstacle avoidance path can be selected under the high dynamic environmental conditions. However, it is also noted from the experiment result that the high nonlinearity of the vehicle model greatly limits the real-time performance and accuracy of the controller’s forward simulation process. Therefore, this controller is more suitable to be used in the situation of relative low vehicle speed rather than a high-speed situation in real-time dynamic path planning.

It can be seen from the abovementioned research that the influence of dynamic model nonlinearity on planner design has increasingly drawn attention, and how to simultaneously improve the planner’s real-time performance and the applied dynamic model completeness should be further studied. Representing a nonlinear vehicle dynamic model by a T–S fuzzy model [20], the path planner can be directly applied to the vehicle model to plan the path with the consideration of the nonlinear vehicle’s dynamic characteristics. In this paper, a Takagi–Sugeno fuzzy-model-based closed-loop rapidly exploring random tree (TS-RRT) planning algorithm is proposed. The nonlinear vehicle dynamics model is described by the T–S fuzzy modelling process. In particular, the dynamic response of the vehicle is precisely described by the T–S fuzzy model for the closed-loop search process and on-line re-planning program. Thus, the optimal local path can be obtained more directly by combining the vehicle dynamics and the environmental information into the path planner. After the uniform boundedness of systems and global asymptotic stability are proven in the controller design process, the steering and braking control output can be calculated by the TS-RRT planner in real-time, and then a proportional–integral–derivative (PID) controller is applied to conduct the trajectory tracking considering the command signal and the vehicle state.

In Section 2, the dynamics model is established and linearized by the T–S fuzzy modelling process. Section 3 presented the closed-loop RRT online planning program and the corresponding trajectory controller design. Section 4 presented the results of simulation evaluation. Finally, the conclusions are drawn in Section 5.

## 2. Modeling of the Vehicle System

The autonomous vehicle is a nonlinear system with strong coupling characteristics and uncertainty. Since the real vehicle dynamics model has high nonlinearity, it is very difficult to apply a linear path-planning algorithm in the practical experiment. For this reason, a T–S modelling method is utilized in this study to build the vehicle dynamic model and maintain the precision of the model. It is noted that this research focuses on the obstacle avoidance path planning in a highly dynamic environment, therefore, the applied dynamic model mainly takes the consideration of the coordinated control system of steering and braking, without a discussion of the driving force input system.

### 2.1. Takagi–Sugeno Fuzzy Model

A simplified three degree of freedom nonlinear vehicle dynamic model is applied in this research, as shown in Figure 1, which can be effectively described by longitudinal speed, lateral speed, and yaw rate. The dynamic model is derived under the following assumptions:
(1)The vertical, roll, and pitch motion is ignored;(2)The braking and steering dynamic are approximated to a linear first-order system;(3)The influence of suspension on tire axle is ignored [21]. The nonlinear model used in our work to represent the vehicle’s dynamics is of the general form as follows:
(1)$$\begin{array}{cc} {\dot{v}}_{x} & =-\frac{c_{x}}{m}v_{x}+\frac{2k_{f}\left(v_{y}+l_{f}\dot{\psi }\right)}{mv_{x}}\delta +\frac{1}{m}a+\tau \left(\Delta x\right),\\ {\dot{v}}_{y} & =-\frac{2\left(k_{f}+k_{r}\right)}{m}v_{y}-\left[v_{x}+\frac{2\left(k_{f}l_{f}-k_{r}l_{r}\right)}{m}\right]\dot{\psi }\\ & \;+\frac{2k_{f}}{m}\delta -\frac{c_{y}}{m}v_{y}+\tau \left(\Delta y\right),\\ \ddot{\psi } & =-\frac{2\left(k_{f}l_{f}^{2}-k_{r}l_{r}^{2}\right)}{I_{z}}\dot{\psi }-\frac{2\left(k_{f}l_{f}-k_{r}l_{r}\right)}{I_{z}}v_{y}\\ & \;+\frac{2k_{f}l_{f}}{I_{z}}\delta +\tau \left(\Delta \psi \right),\\ \dot{\delta } & =\frac{1}{T_{z}}\left(\delta_{c}-\delta \right),\\ \dot{a} & =\frac{1}{T_{a}}\left(a_{c}-a\right), \end{array}$$
where *v_x_*, *v_y_*, and *ψ* represent the longitudinal speed, lateral speed, and yaw angle, respectively; *m* represents the total mass of the vehicle; *I_z_* is the yaw inertia; *l_f_* and *l_r_* the distance from the front axle and the rear axle to the center of gravity, respectively; *c_x_* and *c_y_* are the air resistance coefficients of longitudinal movement and lateral movement, respectively; *f_r_* represents rolling resistance coefficient; *k_f_* and *k_r_* are the stiffness of front and rear tires, respectively; *τ*(Δ*x*), *τ*(Δ*y*), and *τ*(Δ*ψ*) represent the external disturbances and uncertainties caused by time-varying parameters and unmodeled dynamics; *T_z_* and *T_a_* are the first-order hysteretic quantities of the relevant control output reference quantities; *δ_c_* is the control input of steering angle; *a_c_* is the acceleration control input of vehicle body; *δ* is the actual steering angle of the vehicle; *a* is the actual longitudinal acceleration of the vehicle body, and the input signal constraints are
(2)$$\begin{array}{l} a_{\min}<a<a_{\max},\\ \delta_{\max}\geq \left\|\delta \right\|,\\ {\dot{\delta }}_{\max}\geq \left\|\dot{\delta }\right\|, \end{array}$$
where *a*_min_ and *a*_max_ are the maximum and minimum acceleration of vehicle body, respectively; *δ*_max_ represents the maximum steering angle of vehicle; and ${\dot{\delta }}_{\max}$ represents the maximum slew rate for steering.

It is noted that the vehicle dynamics model in (1) is based on the assumption that the uncertainty and external disturbance of the model are limited in a certain range. Under this assumption, we have the continuous function ∏*τ_i_*(*i* = 1,2,3) satisfying the following inequality conditions
(3)$$\begin{array}{c} \tau \left(\Delta x\right)\leq {\bar{\tau }}_{1}\left(v_{x},v_{y},\dot{\psi }\right),\\ \tau \left(\Delta y\right)\leq {\bar{\tau }}_{2}\left(v_{x},v_{y},\dot{\psi }\right),\\ \tau \left(\Delta \dot{\psi }\right)\leq {\bar{\tau }}_{3}\left(v_{x},v_{y},\dot{\psi }\right). \end{array}$$

For the process of T–S fuzzy modelling, the state variables of the vehicle dynamics model can be defined as follows:
(4)$$\begin{array}{ccccc} x_{1}=v_{x}, & x_{2}=v_{y}, & x_{3}=\dot{\psi }, & x_{4}=\delta , & x_{5}=a, \end{array}$$
and the state vector as:
(5)$$x={\left[\begin{array}{ccccc} x_{1} & x_{2} & x_{3} & x_{4} & x_{5} \end{array}\right]}^{T}.$$

We can write a state-space for system (1) as:
(6)$$\dot{x}=Ax+B_{1}w+B_{2}u,$$
where *w* is the external disturbance of the vehicle, *u* is the input of the control system, and
$$w={\left[\begin{array}{ccc} w_{1} & w_{2} & w_{3} \end{array}\right]}^{T},u={\left[\begin{array}{cc} u_{1} & u_{2} \end{array}\right]}^{T}$$
where *w*_1_ = *τ*(Δ*x*), *w*_2_ = *τ*(Δ*y*), $w_{3}=\tau \left(\Delta \dot{\psi }\right)$, *u*_1_ = *δ_c_* and *u*_2_ = *a_c_*. Adding the state to Equation (1), we obtain
(7)$$\begin{array}{cc} {\dot{x}}_{1} & =-\frac{c_{x}}{m}x_{1}+\frac{2k_{f}\left(x_{2}+l_{f}x_{3}\right)}{mx_{1}}x_{4}+\frac{1}{m}x_{5}+w_{1},\\ {\dot{x}}_{2} & =-\frac{2\left(k_{f}+k_{r}\right)}{m}x_{2}-\left[x_{1}+\frac{2\left(k_{f}l_{f}-k_{r}l_{r}\right)}{m}\right]x_{3}\\ & \;+\frac{2k_{f}}{m}x_{4}-\frac{c_{y}}{m}x_{2}+w_{2},\\ {\dot{x}}_{3} & =-\frac{2\left(k_{f}l_{f}^{2}-k_{r}l_{r}^{2}\right)}{I_{z}}x_{3}-\frac{2\left(k_{f}l_{f}-k_{r}l_{r}\right)}{I_{z}}x_{2}\\ & \;+\frac{2k_{f}l_{f}}{I_{z}}x_{4}+w_{3},\\ {\dot{x}}_{4} & =\frac{1}{T_{z}}\left(u_{1}-x_{4}\right),\\ {\dot{x}}_{5} & =\frac{1}{T_{a}}\left(u_{2}-x_{5}\right). \end{array}$$

In order to simplify the high nonlinearity of the proposed model, *f*_1_ and *f*_2_ are defined as:(8)$$\begin{array}{c} f_{1}=\frac{2k_{f}\left(x_{2}+l_{f}x_{3}\right)}{mx_{1}},\\ f_{2}=x_{1}+\frac{2\left(k_{f}l_{f}-k_{r}l_{r}\right)}{m}. \end{array}$$

Then, Equation (7) can be rewritten as
(9)$$\begin{array}{cc} {\dot{x}}_{1} & =-\frac{c_{x}}{m}x_{1}+f_{1}\cdot x_{4}+\frac{1}{m}x_{5}+w_{1},\\ {\dot{x}}_{2} & =-\frac{c_{y}+2\left(k_{f}+k_{r}\right)}{m}x_{2}-f_{2}\cdot x_{3}+\frac{2k_{f}}{m}x_{4}+w_{2},\\ {\dot{x}}_{3} & =-\frac{2\left(k_{f}l_{f}-k_{r}l_{r}\right)}{I_{z}}x_{2}-\frac{2\left(k_{f}l_{f}^{2}-k_{r}l_{r}^{2}\right)}{I_{z}}x_{3}\\ & \;+\frac{2k_{f}l_{f}}{I_{z}}x_{4}+w_{3},\\ {\dot{x}}_{4} & =\frac{1}{T_{z}}\left(-x_{4}+u_{1}\right),\\ {\dot{x}}_{5} & =\frac{1}{T_{a}}\left(-x_{5}+u_{2}\right). \end{array}$$

Therefore, the matrices *A*, *B*_1_
*B*_2_ can be written as
$$\begin{array}{l} A_{\left(5\times 5\right)}=\\ \left[\begin{array}{ccccc} -\frac{c_{x}}{m} & 0 & 0 & f_{1} & \frac{1}{m}\\ 0 & -\frac{c_{y}+2\left(k_{f}+k_{r}\right)}{m} & -f_{2} & \frac{2k_{f}}{m} & 0\\ 0 & -\frac{2\left(k_{f}l_{f}-k_{r}l_{r}\right)}{I_{z}} & -\frac{2\left(k_{f}l_{f}^{2}-k_{r}l_{r}^{2}\right)}{I_{z}} & \frac{2k_{f}l_{f}}{I_{z}} & 0\\ 0 & 0 & 0 & -\frac{1}{T_{z}} & 0\\ 0 & 0 & 0 & 0 & \frac{1}{T_{a}} \end{array}\right],\\ B_{1\left(5\times 3\right)}={\left[\begin{array}{lllll} 1 & 0 & 0 & 0 & 0\\ 0 & 1 & 0 & 0 & 0\\ 0 & 0 & 1 & 0 & 0 \end{array}\right]}^{T},\;B_{2\left(5\times 2\right)}={\left[\begin{array}{lllll} 0 & 0 & 0 & \frac{1}{T_{z}} & 0\\ 0 & 0 & 0 & 0 & \frac{1}{T_{a}} \end{array}\right]}^{T}. \end{array}$$

The T–S fuzzy modelling method is applied here to approximate the high nonlinearity of the system (6). It is noted that the state variables *x*_1_, *x*_2_, and *x*_3_ are actually limited for a stable system, and that the nonlinear *f*_1_ and *f*_2_ should also be bounded. We represent *f*_1_ and *f*_2_ using their minimum values and maximum values by following a “sector nonlinearity” approach:
(10)$$\begin{array}{c} f_{1}=M_{1}f_{1\max}+M_{2}f_{1\min},\\ f_{2}=N_{1}f_{2\max}+N_{2}f_{2\min}, \end{array}$$
where *f*_(*i*)*max*_ (*i* = 1, 2) represents the maximum values and *f*_(*i*)*min*_ (*i* = 1, 2) is the minimum value of the nonlinear *f*_(*i*)_ (*i* = 1, 2). *M*_(*i*)_ and *N*_(*i*)_ (*i* = 1, 2) are fuzzy membership functions and satisfy:
(11)$$\begin{array}{c} M_{1}+M_{2}=1,\\ N_{1}+N_{2}=1, \end{array}$$
and the member functions are defined as:
(12)$$\begin{array}{cc} M_{1}=\frac{f_{1}-f_{1\min}}{f_{1\max}-f_{1\min}}, & M_{2}=\frac{f_{1\max}-f_{1}}{f_{1\max}-f_{1\min}},\\ N_{1}=\frac{f_{2}-f_{2\min}}{f_{2\max}-f_{2\min}}, & N_{2}=\frac{f_{2\max}-f_{2}}{f_{2\max}-f_{2\min}}. \end{array}$$

Then, the nonlinear vehicle model system can be described by the above linear subsystems. For each possibility, there is a corresponding state–space equation:
$$\begin{array}{l} \mathrm{if}f_{1}=M_{1},\;f_{2}=N_{1},\;\mathrm{then}\;\dot{x}=A_{\left(1\right)}x+B_{1}w+B_{2}u,\\ \mathrm{if}f_{1}=M_{1},\;f_{2}=N_{2},\;\mathrm{then}\;\dot{x}=A_{\left(2\right)}x+B_{1}w+B_{2}u,\\ \mathrm{if}f_{1}=M_{2},\;f_{2}=N_{1},\;\mathrm{then}\;\dot{x}=A_{\left(3\right)}x+B_{1}w+B_{2}u,\\ \mathrm{if}f_{1}=M_{2},\;f_{2}=N_{2},\;\mathrm{then}\;\dot{x}=A_{\left(4\right)}x+B_{1}w+B_{2}u, \end{array}$$
where *A*_(*i*)_(*i* = 1, 2, 3, 4). (*i* = 1, 2, 3, 4) are obtained by replacing *f*_(*i*)_(*i* = 1, 2) in matrix A of (6) with *f*_(*i*)*max*_ and *f*_(*i*)*min*_. Then, the T–S fuzzy model for the nonlinear vehicle model under the bounded state variables is obtained as:
(13)$$\begin{array}{cc} \dot{x} & =\sum_{i=1}^{4}h_{i}\left[A_{\left(i\right)}x+B_{1}w+B_{2}u\right]\\ & =A_{h}x+B_{1}w+B_{2}u, \end{array}$$
where $A_{h}=\sum_{i=1}^{4}h_{i}\cdot A_{i},$ and *h*_1_ = *M*_1_*N*_1_, *h*_2_ = *M*_1_*N*_2_, *h*_3_ = *M*_2_*N*_1_, *h*_4_ = *M*_2_*N*_2_, and *h*_(*i*)_(*i* = 1, 2, 3, 4) satisfy $\Sigma_{i=1}^{4}h_{1}=1$.

### 2.2. State Observer

In practice, not all the state variables are available to be measured in real-time. In particular, the velocities are nearly unmeasurable since the direct integration from acceleration to estimate the vertical velocities means the accuracy of the estimation deteriorates as a consequence. To meet input requirements, a planner must be constructed using the estimated state variables and premise variables; that is, to estimate the state variables in real-time, a state observer is designed and integrated with the planner. In terms of the vehicle model, both the steering angle, *δ*, and the longitudinal acceleration of vehicle body, *a*, can be measured by sensors. Therefore, the observer measurement is defined as:
(14)$$\begin{array}{cc} Y & ={\left[\begin{array}{cc} \delta & a \end{array}\right]}^{T}\\ & =C_{1}x, \end{array}$$
where
$$C_{1}=\left[\begin{array}{ccccc} 0 & 0 & 0 & 1 & 0\\ 0 & 0 & 0 & 0 & 1 \end{array}\right].$$

To effectively estimate the state by using the easily measured signals, the estimation error can be defined based on the observer measurement as
(15)$$e=x-\hat{x}.$$

Therefore, the state observer can be designed as (16)$$\begin{array}{cc} \dot{\hat{x}} & =\sum_{i=1}^{4}h_{i}\left[A_{\left(i\right)}\hat{x}+L\left(Y-\hat{Y}\right)+B_{2}u\right]\\ & =A_{h}\hat{x}+L\left(Y-\hat{Y}\right)+B_{2}u, \end{array}$$
where *L* are the state observer gains to be designed. Rearrangement of (16) provides:
(17)$$\dot{\hat{x}}=\left(A_{h}-LC_{1}\right)\hat{x}+LY+B_{2}u.$$

In Equation (15), the error vector of real and estimated states is defined as $e=x-\hat{x}.$, then the dynamic behavior of error can be deduced as:(18)$$\begin{array}{cc} \dot{e} & =\dot{x}-\dot{\hat{x}}\\ & =A_{h}x+B_{1}w+B_{2}u-\left(A_{h}-LC_{1}\right)\hat{x}-B_{2}u-LY\\ & =A_{h}x-(A_{h}-LC_{1})\hat{x}-LC_{1}x+B_{1}w\\ & =\left(A_{h}-LC_{1}\right)e+B_{1}w. \end{array}$$

Here, we have two assumptions that the external disturbance *w* is a Gaussian white noise, whose mean value tends to be zero, and the values of $\left(A_{h}^{T},C_{1}^{T}\right)$ are fully controllable and fully measurable. In system (18), the attenuation of the error is determined by the poles’ locations of the matrix (*A_h_* − *LC*_1_). For a known system, *A_h_* and *C*_1_ are determined by the system characteristics. Therefore, the gain matrix *L* of the designed observer should be chosen to maintain the stability of the system, which is degenerated into a pole assignment problem. Specifically, if the matrix (*A_h_* − *LC*_1_) has an appropriate eigenvalue, the error of the system has a certain decay rate to make the system stable. We have
$$\det\left[\lambda I-\left(A_{h}-LC_{1}\right)\right]=\det\left[\lambda I-{\left(A_{h}-LC_{1}\right)}^{T}\right]=\det\left[\lambda I-\left(A_{h}^{T}-L^{T}C_{1}\right)\right]$$
and since the object control model has been linearized by the Takagi–Sugeno modelling process, the observer design problem can be transformed into a pole assignment problem of $\left(A_{h}^{T},C_{1}^{T}\right)$. To set $L={\left[\begin{array}{ccccc} 0 & 0 & 0 & l_{1} & 0\\ 0 & 0 & 0 & 0 & l_{2} \end{array}\right]}^{T}$, then (*A_h_* − *LC*_1_) can be represented as
$$\begin{array}{l} \left(A_{h}-LC_{1}\right)=\\ \left[\begin{array}{ccccc} -\frac{c_{x}}{m} & 0 & 0 & f_{1} & \frac{1}{m}\\ 0 & -\frac{c_{y}+2\left(k_{f}+k_{r}\right)}{m} & -f_{2} & \frac{2k_{f}}{m} & 0\\ 0 & -\frac{2\left(k_{f}l_{f}-k_{r}l_{r}\right)}{I_{z}} & -\frac{2\left(k_{f}l_{f}^{2}-k_{r}l_{r}^{2}\right)}{I_{z}} & \frac{2k_{f}l_{f}}{I_{z}} & 0\\ 0 & 0 & 0 & -\frac{1}{T_{z}}+l_{1} & 0\\ 0 & 0 & 0 & 0 & \frac{1}{T_{a}}+l_{2} \end{array}\right]. \end{array}$$

In order to ensure that the system is approaching zero, the eigenvalue set *λ* of the system need to all be negative values. Then, the corresponding matrix *L* content (*l*_1_ and *l*_2_) can be calculated to complete the observer design. Thus, the T–S fuzzy model and the corresponding state observer is established and can be applied in the closed-loop RRT planning program.

## 3. Control System Design

To determine the vehicle control input, the existing stochastic planning algorithms generally apply a look-up table to perform the reverse calculation based on the sampled control input value. In this section, the T–S fuzzy-model-based closed-loop RRT algorithm is proposed, which is integrated with a low-order controller to expand the RRT and conduct the on-line re-planning process by considering the closed-loop dynamics. Different from the existing work [15], the proposed TS-RRT samples the input of the stable closed-loop system composed of the controller and the T–S fuzzy dynamic state space. The complete planning and control system is shown in Figure 2.

### 3.1. Takagi–Sugeno Fuzzy-Model-Based Path Planner

In this study, the controller output consists of a series of tuples, which include the steering angle profile of the steering controller and the speed command profile of the speed controller. The TS-RRT uses the controller outputs and the T–S fuzzy dynamics model to conduct a forward simulation process, to calculate the predicted trajectory, and then to check the feasibility of the controller output signal according to the vehicle and environmental boundaries including rollover and obstacle avoidance constraints.

The main idea of this TS-RRT algorithm is to rapidly reduce the distance between a randomly selected node and the tree until all nodes meet the planning requirements. The goal is to find a feasible path from the start point (*x_m_*, *y_m_*) to the end point (*x_goal_*, *y_goal_*). Note that the term *q* show below is equivalent to (*x*, *y*). The exploration process of the RRT planner is shown as follow:
1.Generate a random pot *q_rand_*;2.Find the node *q_nearest_* to *q_rand_* on the tree;3.Connect *q_rand_* and *q_nearest_*;4.Search for nodes on the tree with *q_rand_* as the center and *r_c_* as the radius;5.Find out the potential set of parent nodes *q_parent_potential_*. The purpose is to update *q_rand_* to find if there are any better parent nodes;6.Start to evaluate a random note of potential parent *q_parent_potential_*;7.Calculate the cost of *q_parent_potential_* as a parent node;8.Before the detection of collision, connect *q_parent_potential_* with *q_child_* (that is, *q_rand_*) first;9.Calculate the cost of this path Ω(*t*), *t* ∈ [*t*_1_, *t*_2_];10.Compare the cost of the new path to the cost of the original path. If the cost of the new path is less, conduct the collision detection on it, or it should be replaced by the next potential parent node;11.Collision detection failed, the *q_parent_potential_* is not a new parent;12.Start to evaluate the next potential parent;13.Connect potential parent nodes to *q_child_*;14.Calculate the cost of this path Ω(*t*), *t* ∈ [*t*_3_, *t*_4_];15.Again, compare the cost of the new path to the cost of the original path. If the cost of the new path is less, conduct the collision detection on it, or it should be replaced by the next potential parent node;16.Collision detection passed;17.Delete the previous edge in the tree;18.Add the new edge in the tree, and make *q_parent_potential_* as *q_parent_*.

Thus, the trajectory of a finite period, which depends on the cycle of calculation, can be predicted by the expansion process of the random tree shown above. However, in a dynamic and uncertain environment, trees need to grow continuously in the process of execution, due to the continuously updated vehicle dynamic status and environmental information. Therefore, real-time planning requires the vehicle dynamic model state and the reuse of the information from previous calculation cycles [22,23]. The re-planning program of the TS-RRT planner, which is also demonstrated in Figure 3, is designed as follows:
1.Open the re-planning program;2.Update the current vehicle T–S fuzzy states *x_TS_*(*t*_0_);3.Update the environmental constraints Γ_feasible_ (*t*) from the obstacle configuration space;4.Apply the state observer to propagate the states by a computational time step Δ*t* and obtain *x_TS_*(*t*_0_ + Δ*t*);5.Conduct the random tree exploring process;6.Until calculation time limit Δ*t* is reached;7.If no such sequence exists, then send emergency stop to controller and return to step 2;8.End if;9.Choose the best safe node sequence in the tree;10.Re-propagate the latest T–S fuzzy state *x_TS_*(*t*_0_ + Δ*t*) using the Ω(*t*) with the best node sequence, and then obtain *x*(*t*);11.If *x_TS_*(*t*)∈Γ_feasible_(*t*), then send the best potential path Ω(*t*) to the controller;12.If anything else, delete the previous infeasible path in the tree and return step 9;13.End if;14.Until the vehicle reaches goal.

The algorithm shown above illustrates the program of the TS-RRT algorithm to execute the part of tree exploring and to grow the tree while the controller executes the path planning in real-time. The planner sends the input to the controller at a fixed rate per second and the extension of the tree continues until the time limit (line 7) is reached. After each computation cycle, the best track is selected as the node sequence with control input *U_cmd_* = [*δ_cmd_*   *a_cmd_*]*^T^*. These signals are sent to the controller to be added to the reference path for trajectory control (line 13). 

It is noted that when selecting the best path, only the sequence of nodes that end in a safe state is considered. If not, the planner commands the controller to perform an emergency brake to stop the car as soon as possible for security.

### 3.2. Trajectory Controller Design

During the TS-RRT planner conducting the path planning program, we use a simple and effective PID controller to track the designed trajectory in real-time.

For acceleration tracking, a simple PID controller is considered. However, due to the inherent speed damping of the vehicle and the noise of the acceleration signal, a PID controller has no obvious advantage to a PI controller, which has fewer parameters to be designed. Therefore, a PI controller is applied here and shown as follows:
(19)$$a_{c}=K_{p\left(a\right)}\left(a_{cmd}-a\right)+K_{i\left(a\right)}\int_{0}^{t}\left(a_{cmd}-a\right)dr$$
where u is the dimensionless speed control signal; *K_p_* and *K_i_* are proportional gain and integral gain, respectively.

Similarly, our steering controller is designed as:
(20)$$\delta_{c}=K_{p\left(\delta \right)}\left(\delta_{cmd}-\delta \right)+K_{\delta \left(\delta \right)}\int_{0}^{t}\left(\delta_{cmd}-\delta \right)dr$$

After constructing the planner and the trajectory controller, the T–S fuzzy model is left aside in the simulation work, and the complete closed-loop control system is shown in Figure 2.

For vehicles with complex dynamics, the dimension of vehicle state might be established with a very high dimension. However, the control signal output from the applied PID controller has a lower dimension, which can effectively guarantee the real-time performance of trajectory tracking.

## 4. Numerical Evaluation

In the section, an experiment in the loop simulation was conducted with a 1000 Hz real-time simulation frequency to verify the algorithm’s effectiveness. The nonlinear vehicle dynamic model, considering the dug-off tire model [24], is applied as the plant model. Meanwhile, the designed TS-RRT planner runs at 20 Hz based on an AGX Xavier chip with the vehicle model parameters listed in Table 1. To validate the performance of the TS-RRT, a traditional RRT planner is adopted for a comparison. The simulation result in Figure 4 presents the planned path of the autonomous vehicle in the presence of three moving obstacles. The vehicle adjusts its speed and direction when it detects the moving obstacle in its path, and sometimes it slows down when making its decision. It is noted that only the results of TS-RRT are shown in Figure 4, and the results of the competing controller, traditional RRT, are omitted for clarity.

To further illustrate the effectiveness of the TS-RRT algorithm, we constructed two test scenes including a highway case and a parking case, using a traditional artificial potential field (APF) method and an A* algorithm as a competitor for the simulation. The results of several typical scenarios are shown in Figure 5 and Figure 6.

The simulation result in Figure 5 presents the planned paths of the two competing planners, TS-RRT and APF, in the scene of motorway driving. It can be detected from Figure 5a that the APF algorithm has the same planning effect as TS-RRT when the environment complexity is low. However, in a complex and jam-packed environment, as shown in Figure 5b, the potential field force of the APF algorithm is trapped in the dilemma of local optimum, which results in the planning failure and the vehicle’s emergency stop. In Figure 5c, it can be seen that the front car is intending to slow down and merge to the left lane. In this case, with rolling optimization that includes re-planning and stitching, the TS-RRT planner generates a shorter and smoother trajectory than the APF one to complete the overtaking task.

The result in Figure 6 presents the planned paths of the two competing planners, TS-RRT and A*, in the scene of parallel parking. Comparing the results in Figure 6b,c, it can be found that the TS-RRT algorithm, which effectively considers the vehicle kinematics and dynamic characteristics, is appropriate for dealing with the narrow parking scenarios, and can complete the side parallel parking planning without collision. On the other hand, although the A* planner is improved with grid subdivision processing and accessibility path filtering, the final planned path is not available (a collision occurs) since the algorithm is lacking in dynamic characteristics consideration in the situation of tire slipping (simulated by the low road friction coefficient setting) and deceleration strip passing.

In order to illustrate the simulation results more clearly, the performance of the TS-RRT algorithm, the traditional RRT algorithm, and APF algorithm are compared in Table 2 in terms of the total planned distance, the time consumption, and the average vehicle speed. The value of the specific object is the average of ten simulation results. By the comparing the results, it can be seen that the proposed TS-RRT algorithm is optimized in terms of the total distance and the time consumption compared with the competing methods in the dynamic obstacle environment.

Table 3 illustrates the planning results and the time consumption of the two controllers in ten simulations. Under the dynamic obstacle environment, the search speed of TS-RRT is significantly faster compared with the traditional RRT search algorithm, and the search success rate is greatly improved with more propagations.

Furthermore, the observer’s effect that collected the simulation are organized and shown in Figure 7. Specifically, the four subfigures demonstrate the results of the two-observable states, steering angle and longitudinal acceleration, with measured inputs from sensors and the corresponding estimated value from the state observer, respectively. Moreover, the estimated input and output signals of the estimator calculate the delay as around 0.5–1 ms, which could be accepted within a 5 ms calculation period mission.

## 5. Conclusions

In this paper, a nonlinear vehicle dynamic model is considered in the control system design. For local path planning, the T–S fuzzy modelling method is applied to the nonlinear dynamic model to help the path planner to create a feasible path. Then, a closed-loop RRT algorithm with an on-line re-planning process is applied to build the path planner, which effectively improves the vehicle performance of dynamic obstacle avoidance, and plan the local obstacle avoidance path in line with the dynamic characteristics of the vehicle. Finally, the performance of the planner is evaluated by numerical simulation. The results demonstrate that the proposed controller can effectively plan the path and support a favorable trajectory.

## Figures and Tables

**Figure 1 sensors-23-00443-f001:**
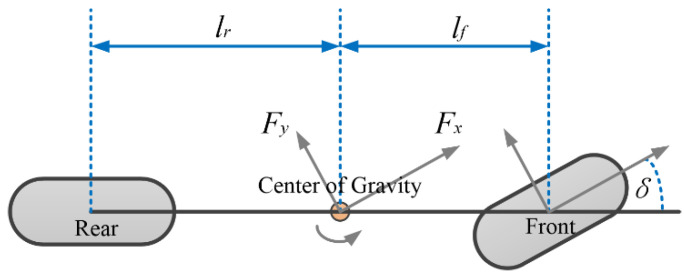
Vehicle dynamics model.

**Figure 2 sensors-23-00443-f002:**
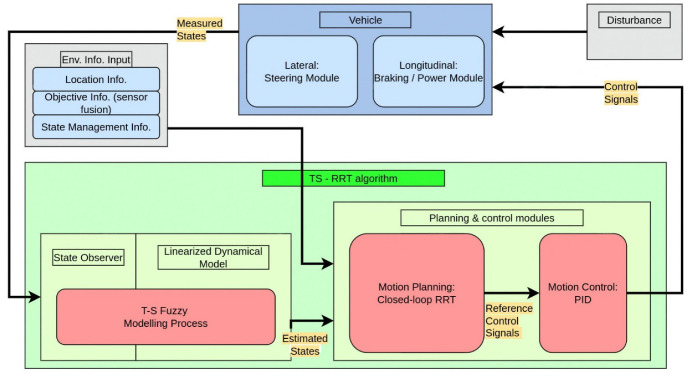
The control flow of the closed-loop system.

**Figure 3 sensors-23-00443-f003:**
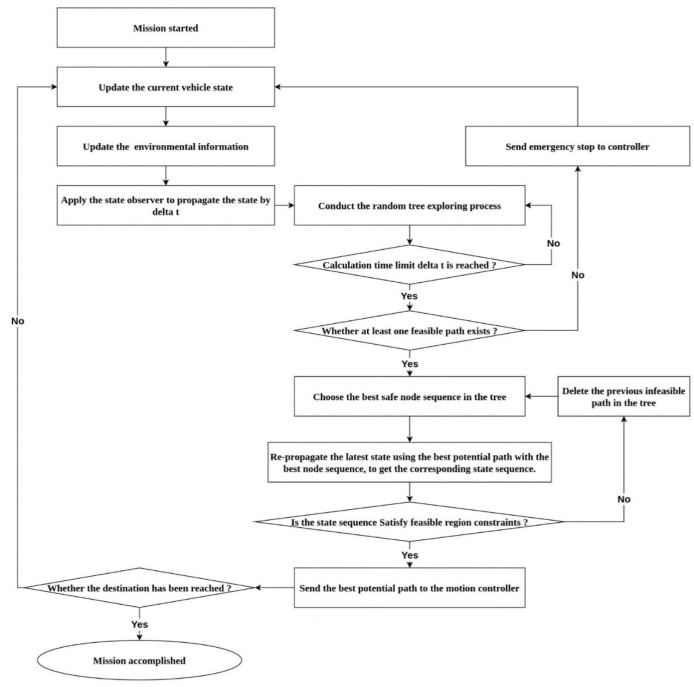
The pipeline of the TS-RRT planner.

**Figure 4 sensors-23-00443-f004:**
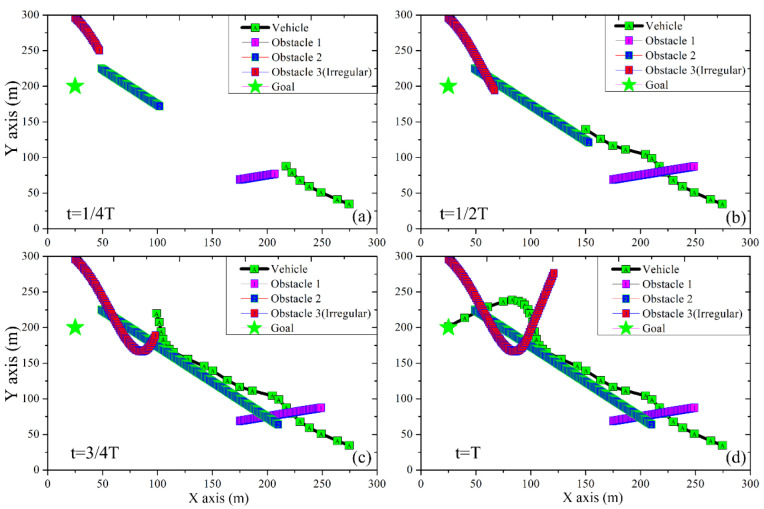
The planning results in dynamic environment: (**a**) at 1/4 of the total time, (**b**) at half of the total time, (**c**) at 3/4 of the total time, (**d**) at the end of the planned period.

**Figure 5 sensors-23-00443-f005:**
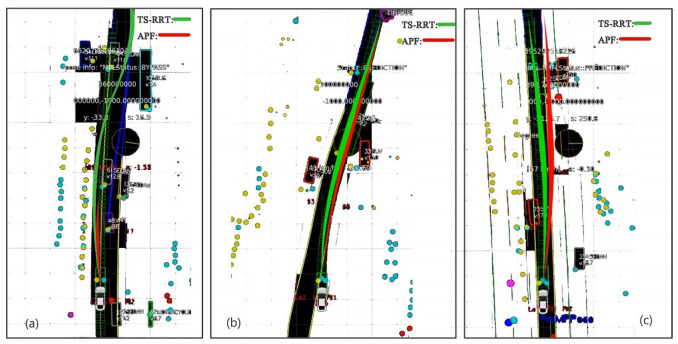
Comparison of algorithm effectiveness under motorway running: (**a**) bypassing in jam-packed case, (**b**) bypassing in normal case, (**c**) overtaking case.

**Figure 6 sensors-23-00443-f006:**
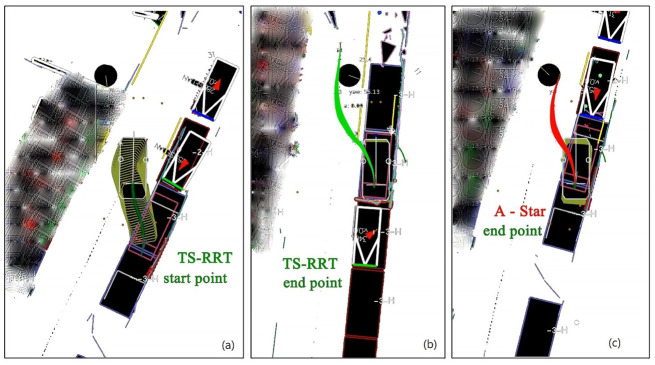
Comparison of algorithm effectiveness in parking planning: (**a**) start status of TS-RRT planning, (**b**) end status of TS-RRT with successful planning, (**c**) end status of A* planning with collision failure.

**Figure 7 sensors-23-00443-f007:**
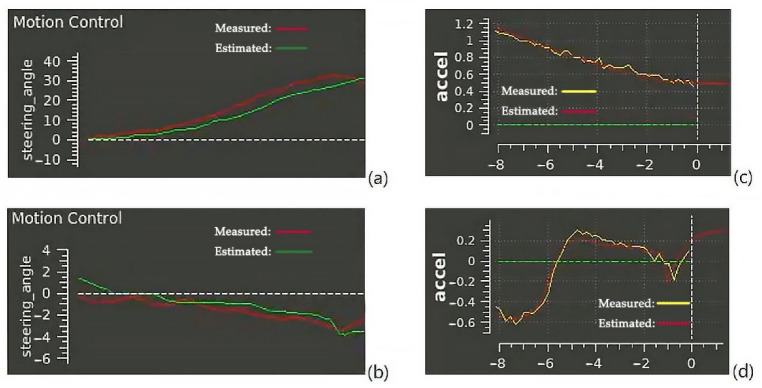
State monitoring and estimation: (**a**) steering angle with large amplitude, (**b**) steering angle with small amplitude, (**c**) longitudinal acceleration with large amplitude, (**d**) longitudinal acceleration with small amplitude.

**Table 1 sensors-23-00443-t001:** Model parameters.

Parameter	Value	Parameter	Value
*δ* _max_	0.539 rad	*m*	1589 kg
${\dot{\delta }}_{\max}$	0.331 rad/s	*I_z_*	36,918 kg·m^2^
*a* _min_	−8.2 m/s^2^	*l_f_*	1.38 m
*a* _max_	3.8 m/s^2^	*l_r_*	1.67 m
*T_a_*	0.35 s	*k_f_*	379 kN/m
*T_z_*	0.35 s	*k_r_*	388 kN/m

**Table 2 sensors-23-00443-t002:** Comparison of results (average value of 10 times).

Strategy	APF Algorithm	Traditional RRT	TS-RRT
Trajectory distance	338.30 m	325.93 m	304.23 m
Time consumption	24.09 s	25.11 s	21.38 s
Average vehicle speed	14.04 m/s	12.98 m/s	14.23 m/s

**Table 3 sensors-23-00443-t003:** Iterations and time consumed to generate the optimal track.

Strategy	Traditional RRT	TS-RRT
Potential trajectory	86	112
Total propagation	11,408	14,253
Time cost	11.85 s	10.24 s
Propagation/cycle	310	402
Time cost/cycle	0.32 s	0.29 s

## Data Availability

Not applicable.

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
