# Peer review of "A Study on Dynamic Motion Planning for Autonomous Vehicles Based on Nonlinear Vehicle Model"

_sensors, 2022, doi:10.3390/s23010443_

Round 1

Reviewer 1 Report

1.      The abstract need to be more concise to show the highlights of this paper.

2.      There are many other path planning algorithms such as A*, potential field, and neural network. Please show more advantages of this method.

3.      Please explain why the states are observable based on the measured variables in equation (14).

4.      The state observer gains in equation (18) are important and need to be designed. Please present the design criterion, the design method, and its actual performance.

5.      The authors can use the linearized model to design the controller. But it is better to utilize a detailed nonlinear model of the vehicle to simulate the system’s dynamic performance.

6.      The simulation result should be compared to other new methods to demonstrate its advantage. Only the traditional RRT planner is not enough.

7.      The simulation frequency 20Hz is too slow for a controller in this application. The author should use a more detailed vehicle dynamic model to verify its performance.

Reviewer 2 Report

This paper focuses on the Dynamic Motion Planning for Autonomous Vehicles Based on Nonlinear Vehicle Model. Some places need to be further clarified:

1. The physical meaning of equation (1) in section 2.1 should be explained in more detail. Furthermore, symbols $T_z$ and $T_a$ are not defined in equation (1).

2. For vehicles with complex dynamics, the dimension of vehicle state might be established with a very high dimension. However, the control signal output from the applied PID controller has a lower dimension, which can effectively guarantee the real-time performance of trajectory tracking. What is the significance of the vehicle state dimension and the output control signal dimension, respectively? 

3. According to system (7), $f_1$ and $f_2$ are definded to handle nonlinear items. It is noted that the state variables x1 , x2 and x3 are actually limited, then the nonlinear $f_1$ and $f_2$ should also be bounded. However, in the actual vehicle experiment, how to get the upper and lower bound of $f_1$ and $f_2$. A detailed explanation should be given. 

4. The overall layout of this article is unacceptable and needs lots of modification. Furthermore, there are some writing mistakes.  For example, ")" in line 62 of the introduction is missing and symbol "->" in line 269 is wrong?

Round 2

Reviewer 1 Report

The authors have answered all my questions. Many thanks. My suggestion now is accepted in present form.

Author Response

Dear reviewer:

Thank you for your comment and kindly help!